# New Ballistic Neutron Guide for the Time-of-Flight Spectrometer FOCUS at PSI

Fanni Juranyi [1,*], Masako Yamada [2,*,†], Christine Klauser [2], Lothar Holitzner [2] and Uwe Filges [2]

1   Laboratory for Neutron Scattering and Imaging, Paul Scherrer Institut, CH-5232 PSI Villigen, Switzerland
2   Laboratory for Neutron and Muon Instrumentation, Paul Scherrer Institut, CH-5232 PSI Villigen, Switzerland; christine.klauser@psi.ch (C.K.); lothar.holitzner@psi.ch (L.H.); uwe.filges@psi.ch (U.F.)
*   Correspondence: fanni.juranyi@psi.ch (F.J.); masako.yamada@kek.jp (M.Y.)
†   Current address: Institute of Materials Structure Science, High Energy Accelerator Research Organization, Tokai, Ibaraki 319-1106, Japan.

**Abstract:** FOCUS is a direct-geometry cold neutron time-of-flight spectrometer at SINQ (PSI, CH). Its neutron guide was exchanged in 2019/2020 within the SINQ Upgrade project, while the rest of the instrument remained unchanged. The new guide provided a significant intensity increase across the whole spectrum, especially at short wavelengths, due to the more efficient transport and extended phase space of the transported neutrons. The practically available energy transfer range (at the neutron energy loss side) was increased to about 40 meV. The main reason for the intensity benefit at short incident wavelengths was the improved guide coating, whereas at long wavelengths it was the new ballistic shape. The interesting part of the guide is the "peanut shape" of the curved part in the horizontal plane. For this, we derived the analytical restriction on the geometry to avoid a direct line of sight from the source. The guide geometry and the supermirror coating were optimized using Mcoptimize, a particle swarm optimization routine employing Mcstas. Future ballistic neutron guides may profit from the presented approaches, optimization strategy, and results.

**Keywords:** neutron guide; ballistic guide; generalized direct line-of-sight condition; particle swarm optimization; Mcstas; neutron spectrometer

## 1. Introduction

FOCUS is a direct-geometry, cold neutron time-of-flight spectrometer at the quasi-continuous source SINQ (PSI, CH) [1]. It is used to study diffusion, phonons, and magnetic excitations in solid or liquid materials. Most experiments use an incident wavelength in the range of $\lambda_i =$ 2–6 Å, i.e., the (002) reflection of the pyrolytic graphite (PG) monochromator. The monochromator is double-focusing (composed of 63 crystal pieces) to maximize the neutron flux at the sample. The PG monochromator can be exchanged in order to cover the whole available incident wavelength range. The second monochromator is made of natural mica crystals ($\lambda_i < 15$ Å); however, the low reflectivity limits its usage severely. The time structure of the neutron beam is shaped by a Fermi chopper, which is located between the monochromator and the sample. The scattered neutrons are detected by 375 $^3$He counters (0.72 sr), which are organized in three banks around the sample at a distance of 2.5 m. See the layout in Figure 1.

The neutron guide of FOCUS, as well as those of all other instruments at the cold neutron source, were exchanged within the SINQ Upgrade project. The technological developments over the 25 years since the original installation offered the opportunity to not simply replace the old neutron guide but to improve the guide with an advanced design.

Several studies on guide geometry optimization have recently been performed because of the construction of new neutron sources and the implementation of numerous instrument upgrade programs. It has been demonstrated that with increasing guide length and beam divergence, a ballistic neutron guide (increased cross-section in the

middle part of the guide) becomes essential, as it minimizes the intensity losses caused by the imperfect reflectivity of the supermirror coating [2,3]. In a ballistic guide, the loss of flux is reduced because of the following reasons [4]: (i) the number of reflections is inversely proportional to the guide cross-section for a given divergence, and (ii) the divergence is reduced according to Liouville's theorem. As a corollary, (ii) implies a higher average reflectivity per reflection, because the reflectivity of a supermirror-coated guide depends on the scattering angle. In the first approximation, the reflectivity of a supermirror can be described with the two main parameters $m$ and $\alpha$. Below m = 1 (Q $\approx$ 0.02 Å$^{-1}$), the imperfect total reflection is estimated as 99.5%. At higher angles (or momentum transfers), the reflectivity decreases with a slope of $\alpha$ until m = $m$, where $m$ denotes the critical angle of reflection relative to that of Ni. Above $m$, the reflectivity is practically zero.

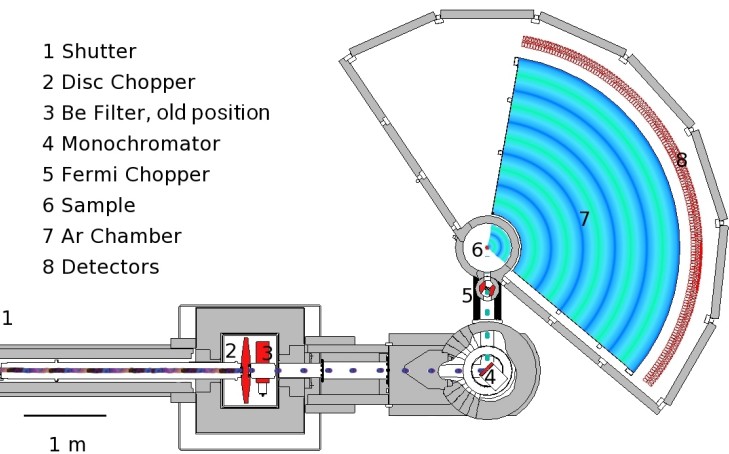

1 Shutter
2 Disc Chopper
3 Be Filter, old position
4 Monochromator
5 Fermi Chopper
6 Sample
7 Ar Chamber
8 Detectors

1 m

**Figure 1.** Layout of the time-of-flight spectrometer FOCUS.

Klenø et al. compared the performance gains of elliptic, parabolic-straight-parabolic, and tapered-straight-tapered guides over a straight guide for different guide lengths, divergences, and wavelength ranges [5]. For the conditions at FOCUS (guide length of 68 m, with a primary interest in the wavelength range of 2–6 Å and a maximum divergence of more than 1°), they reported similar intensity gains for the first two cases, about a factor two more than for the tapered-straight-tapered guide. Note that the authors used the term "ballistic" only for a tapered-straight-tapered combination in [5], while we use it in a more general way, also including an elliptical shape, similar to [3]. An elliptic guide was one of the first advanced guide shapes considered, because of its distinct focusing behavior: a point source can be imaged exactly, and each trajectory involves no more than one reflection. Although this is not true for a finite real source size [6], it is still one of the best-performing geometries to date.

Geometrical constraints also have to be taken into account when designing a guide: most importantly, the guide needs to avoid the direct line of sight (LOS) of the source to minimize the fast neutron background in the neutron guide hall. This can be achieved by changing the guide direction via either a curved section, a kink, or placing shielding elements in the middle of the guide. Such measures result in a significant intensity reduction, even if the components are optimized [7]. If the guide is composed of different shapes, the connection should be smooth to avoid significant gaps in the divergence profile [8]. Further geometrical limitations arise from restrictions imposed by the layout of the experimental hall as well as integrated components like choppers, filters, and shutters. These might require the guide cross-section to be reduced locally.

In this paper, we describe the design concept, implementation, and performance improvement of the new neutron guide for FOCUS.

## 2. The Original Guide

The FOCUS guide is 68 m long, and its path (central line) was not changed during the upgrade, in order to comply with the geometrical restrictions. It starts 1.5 m behind the cold source with a 4.6 m long straight insert across the shielding wall, which has strict geometrical limitations. After this, there is a 20 m long curved section with a bending radius, R = 1445 m, to avoid direct LOS. The path of the last straight section is 43.3 m long.

The old guide width was 5 cm throughout. The overall guide height was 12 cm, apart from the last 2.95 m, where the beam was compressed to 9.5 cm with a linearly tapered section in order to comply with the disc chopper geometry. This last section had an m = 3 supermirror coating on the top and bottom, with m = 2.4 on the sides, while the rest of the guide had m = 2 on all sides.

## 3. The Design Concept of the New Guide

We aimed to maximize the neutron flux (intensity) at the sample while preserving the energy resolution, which is the most important parameter for an experiment on a time-of-flight spectrometer. Optimally, the different contributions to the energy resolution should be matched. A new guide might affect the contribution from the distribution of the scattering angles on the monochromator, i.e., the angular view of the guide end from the monochromator crystals [1]. Notably, the divergence delivered by the guide system does not affect the minimum of the energy resolution as a function of energy transfer. Instead, a high divergence provides illumination to a greater area of the monochromator, which determines the wavelength distribution across the monochromator and thus influences the energy transfer dependence of the energy resolution [1]. The angular (momentum transfer) resolution on a time-of-flight spectrometer has a secondary priority. It is often relaxed to increase the phase space of the incoming neutrons, i.e., their intensity. At FOCUS, it is determined by the size of the monochromator.

Therefore, we kept the end dimensions of the guide, the size of the monochromator crystals, and the area of the whole monochromator and thus preserved the instrument characteristics. Since we only changed the guide, the intensity gain at the sample would be roughly the same as the gain at the monochromator. We also would like to remark that the beam profile at the sample position was determined by the mosaicity (0.8°) and the size of the monochromator crystals. This meant that the intensity gain did not depend on the sample size.

For the design of the new guide, we used the following boundary conditions:

- The instrument had to be out of the line of sight (LOS).
- The instrument stayed at the same position (the guide path was preserved for the final concept, which meant that only the guide shape and coating changed with respect to the old guide). Relocation of the instrument was not a feasible option within our boundary conditions.
- the space is restricted at the first 4.6 m of the guide (in the insert).
- The guide exit size was fixed by the resolution requirements (and limited by geometrical constraints, such as the disc chopper window size).
- The guide should be evacuated (without an additional vacuum box; max. width and height ≤ 250 mm). Early simulations showed that the achievable benefit did not justify the extra costs.
- The *m*-value of the coating for the straight insert close to the cold source was limited to 4.5, which we thought would withstand radiation well enough in this highly irradiated area. For the rest of the guide, the *m* value was limited to 7, which was the highest available value at the time of the design.
- Two gaps with suitable dimensions had to be introduced for two shutters and a Beryllium filter (their locations had some flexibility).

The guide upgrade was a unique opportunity for further changes, like the relocation or addition of devices. Ideally, a velocity selector should be placed close to the source (within the well-shielded bunker) to minimize the background by limiting the wavelength

range of the transported neutrons. Additionally, this solution would prevent higher-order contamination as well. Unfortunately, this was not compatible with the ballistic shape of the guide (increased cross-section).

The chopper system, as it was, could be used to eliminate the higher-order reflections (n $\lambda$) from the monochromator in some but not all cases; therefore, we relied on a Beryllium filter. Originally, the filter was located at the end of the guide, between the two choppers and in the proximity of the sample space, where it created a significant amount of background. Therefore, in the new design, the filter was relocated to the first gap of the guide, shortly after the curved section inside the well-shielded bunker, far away from the FOCUS spectrometer and other instruments. A new filter design was needed because of the change in the cross-section. Experience from PSI and ISIS [9] showed that shortening the filter from the original length of 17 cm to 10 cm would be sufficient to eliminate higher-order peak(s) from the spectrum, while we could benefit from the higher transmission of the desired neutrons. Beyond 10 cm in length, multiple scattering can play an important role, and it can be more efficient to use neutron-adsorbing sheets within the Be block instead of further increasing the length [9]. The new filter was designed accordingly.

Finally, we would like to point out that although only the guide was upgraded, the new design did not limit further instrument development—quite the opposite, the new design opened up new instrument development possibilities.

## 4. Methods and Materials

We used state-of-the-art ray-tracing simulation tools that were able to capture the complexity of the neutron beam and the components: McStas [10], Version 2.1. at that time; Guide_bot [11]; and mcoptimize [12]. The latter is not (yet) well known in the community. It is a particle swarm optimization routine written in Matlab/Octave. It was kindly provided by M. Appel (ILL), who used the same software for optimizing the adaptive section of the IN16B guide, needed for the BATS mode [13]. Similar kinds of optimization had already been used for neutron optics [14]. First, Guide_bot was used to obtain a quick overview of the possible gain factors and performance of different guide geometries (various combinations of linear, curved, elliptic, and parabolic shapes, including kinks). The optimization of the guide geometry and the coating was then finally carried out using mcoptimize. The source spectrum was adapted to match experimental values obtained at the ICON beamline [15]. We used the Guide_gravity and Elliptic_guide_gravity components in McStas to consider the effect of gravity, which influenced the neutron path and, indirectly, the efficiency of the transport.

The guide substrate in the insert was float glass without boron, with an overall thickness of 10 mm. For the rest of the guide, polished borkron glass was used, with a thickness of 15 and 25 mm in the case of top/bottom and side pieces, respectively. The insert (as part of a bundle of three guides) was constructed and built by Mirrotron Ltd. (Budapest, Hungary). The main part of the new guide was built by Swiss Neutronics AG, but the guide pieces were coated mainly in house in order to stay within budget. For each sputtering run, a smaller and thinner (2 and/or 10 mm) test substrate made of either float glass or borkron was coated together with the guide pieces using the same target-to-substrate distance. The reflectivity of each test piece was measured using various instruments: CRISP at ISIS, D17 at ILL, and Hermes at LLB.

The simulation results were compared to the measurements. First, the simulation of the original guide and the spectrometer was thoroughly validated based on the measured time-of-flight spectra before and after the Fermi chopper for various instrument setups. In the end, the performance gain with the new guide was evaluated in different ways. The white beam flux was measured by gold foil activation, which was performed before and after the upgrade directly behind the insert (at 6.1 m from the source) and at the end of the guide. With this method, the beam flux integrated over the whole spectrum was measured over an area of $\varnothing = 1$ cm at the center of the guide cross-section. The flux calculated from the gold foil activation measurements depended on the capture reaction rate (the

convolution between the $^{197}$Au cross-section and spectrum). Since the spectrum is not well known, we only present the ratios of activities. The intensity distribution at the guide end was measured by a dysprosium sheet in order to estimate the neutron current gain in the guide. Additionally, the white beam current at the guide end was also measured with our regular fission chamber monitor. For the monochromatic neutron beam, the intensity gain was measured: (i) at the sample position by gold foil activation over an area of $\varnothing$= 1 cm; (ii) close to the sample position by a well-shielded $^3$He detector behind a $\varnothing$= 1 mm pinhole; (iii) by another fission chamber monitor in front of the sample, covering the whole area of the neutron beam; and (iv) by the intensity of the elastically scattered neutrons using a vanadium standard sample. Lastly, the full time-of-flight spectra of the vanadium standard were compared with a representative setup, $\lambda_i$ = 4.3 Å in order to compare the energy resolution and the background.

Note that the neutron flux changes were not entirely due to the guide upgrade. The SINQ target and proton beam conditions were different before and after the upgrade, which affected the measured neutron fluxes. These additional factors were corrected to the best of our knowledge by the reference measurement at the BOA beamline. BOA employs the same cold source as FOCUS, but it is located on the other side of SINQ, and the instrument stayed the same before and after the upgrade. Thus, we considered the change in BOA flux ($\approx$30%) as the effect of these additional changes to calculate the effect of the new guide of FOCUS.

## 5. Results

### 5.1. Upgrade Potential

The instrument was originally coded in McStas and validated by experiments in 2004 within the 6th EU framework program NMI3. After the revision and adaptation to McStas Version 2.1, the whole instrument was examined, and the upgrade possibilities were identified and quantified. In the following, we focus the discussion on the replacement of the neutron guide. We evaluated the performance of the old guide (beam divergence and distribution) at the monochromator position, which helped us to recognize the opportunities. The characteristic wavelength range used at FOCUS is represented by the two incoming neutron wavelengths, 2 and 6 Å, which are the limits provided by the (002) reflection of the PG monochromator.

Figure 2b,d show the divergences at the end of the old guide. The divergence transported by the guide is limited by the critical angle of the supermirror reflectivity, which is approximately $\theta_c = 0.099\, m\, \lambda = 0.4$ and $1.2°$ [16] for $\lambda = 2$ and 6 Å, respectively. This is why the transported divergence is much larger for longer wavelengths in general. According to the simulations, the transported divergence was smaller than the previously calculated critical angles for both cases. A high *m* value allows for a greater phase space to be transported, but only if the transport is efficient. For a straight guide with a constant width ($w$), the neutron transmission (in 2D only) depends on the incident angle ($\psi$) as follows: $T = \left(R_0 - \frac{\phi}{0.1\lambda}\alpha\right)^{tan(\psi)l/w}$, where $R_0$ is the total reflectivity, close to 1.0, and $l$ is the guide length. Taking state-of-the-art reflectivity values, we realized that the transmission of 6 Å neutrons was very small already for a $1°$ scattering angle. In agreement with this, the simulations predicted a significant intensity gain if we changed the coating of the old guide from m = 2 to m = 3 (a factor of approximately 3), but not much beyond that (less than a factor of 4 for the highest possible coating). This meant that the efficient transport of the maximum possible divergence could only be achieved by an additional reduction in the number of reflections (and hence the losses in the guide) using a ballistic geometry.

As discussed in Section 2, the transported divergence affects the area of illumination at the monochromator, which we study next. The intensity distribution at the monochromator for the original instrument was different for the two wavelengths, see Figure 2a,c: the height of the illuminated area was greater when the transported divergence was higher (as expected), but its width had the opposite trend. To understand this, we had to consider that the monochromator was set at $\theta =17.34$ and $63.4°$ for 2 and 6 Å, relative to the incoming

neutron beam, respectively. The corresponding projected width of the monochromator was 53 and 161 mm, respectively. We see that in the original guide design, the monochromator was not fully illuminated. Hence, increasing the intensity by increasing the transported divergence and the transport efficiency was the goal of the new design.

Raising the intensity on the monochromator by increasing the transported divergence would result in a significant intensity increase at the sample position. In the vertical direction, this would be a pure gain without any penalty. In the horizontal direction, this would have a small effect on the energy resolution, but only for large energy transfers, because FOCUS is operated in the time-focusing mode [1].

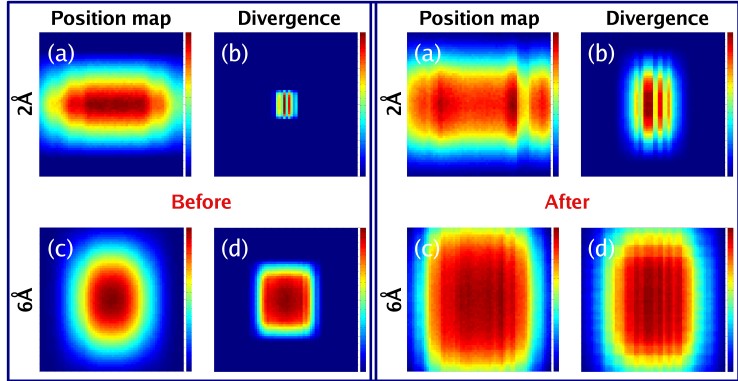

**Figure 2.** Simulated intensity distribution at the monochromator surface ((**a**,**c**); the panels show the area of the monochromator, $180 \times 180$ mm$^2$) and the divergence ((**b**,**d**); the panels show the ranges between $\pm 2°$) at the end of the neutron guide for 2 (top row) and 6 Å (bottom row). The performance of the guide before and after the upgrade is compared. The new guide illuminated the monochromator much better, and the instrument could profit from the significantly increased divergence. The key changes were the improved supermirror coating and the reduced number of reflections due to the ballistic shape.

### 5.2. Direct Line-of-Sight Condition

Because of the fairly large guide cross-section and the requirement to preserve the original curved guide path, we excluded the option of using a shielding element in the center of the guide. Instead, we derived a generalized LOS condition analytically and used that in the parametrization of the guide system for the Mcoptimise simulations. Please note that Guidebot could not handle the LOS request for a fixed guide path.

A curved guide with radius $R$ and constant width $w$ has to be at least $L = \sqrt{8wR}$ long in order to avoid the direct line of sight; see, e.g., [17]. One can derive this equation based on three points, which give the limiting condition for having a straight trajectory between the start and the end of the LOS section. At the end of the LOS section, neutrons that do not interact with the guide coating leave the guide. It is therefore advisable to design the guide so that the end of the LOS section is within a well-shielded area in order to suppress the high-energy neutron background. In our case, we wanted the LOS to end in the neutron guide bunker, at the end of the curved section.

In the following, we derive the analytical LOS condition for our guide geometry (applied in the horizontal plane) by generalizing the above case of a simple curved guide. In the case of the simple curved guide, only three points matter, where the straight trajectory touches the guide: $A$, the outer wall at the entrance; $C$, the outer point at the end of the LOS section; and $B$, where the connecting line touches the inner wall, which is at half of the distance for symmetry reasons. The same equation also holds for many other guide shapes, as long as these three points are fixed. We went a step further by releasing the requirement of a constant guide width and by adding a linear part with a length of $d_i = 4.6$ m at the beginning of the guide (the insert section) to comply with our guide path.

The three points, $A$, $B$, and $C$, had to fall on a straight line (Figure 3). $A$, which is the outer side at the entrance, was fixed in our coordinate system by the boundary conditions.

*C* is a point at the end of the curved section, on the outer side. Its coordinates depended only on the guide width, $w_o$; all other parameters were determined by the fixed guide path. The third point is *B*, located somewhere between *A* and *C*, on the inner side of the guide. The most convenient approach was to parameterize its coordinates by the guide width, $w_m$, and by the angle $2\phi$, which corresponds to an arc of the curved section.

The three points fall onto a straight line if Equation (1) is fulfilled. Or, expressed with the coordinates, we required the tangents of these angles to be equal and obtained Equation (2).

$$\angle OAB = \angle OAC \tag{1}$$

$$\frac{R\left(1 - \cos 2\phi\right) + w_m/2 + w_i/2}{R\sin 2\phi + d_i} = \frac{R\left(1 - \cos 2\alpha\right) - w_o/2 + w_i/2}{R\sin 2\alpha + d_i}, \tag{2}$$

where $2\phi$, $w_m$, and $w_o$ are the unknown variables, described above; *R* is the radius of the curved section; $w_i$ is the guide width at the start; $d_i$ is the length of the straight part (insert); and $2\alpha$ corresponds to the arc of the curved section. Finally, we introduce *O* as an arbitrary point on the horizontal line starting at *A*, i.e. having coordinates of [x,w/2], which is just simplifying the definition of the angles. We could eliminate one of the three variables by recognizing that (for any given *C*, i.e., the right-hand side of Equation (2) is a constant) the guide width at point *B* has to be the maximum value (Equation (3)).

$$\frac{\partial \tan(\angle OAB)}{\partial(2\phi)} = 0 \tag{3}$$

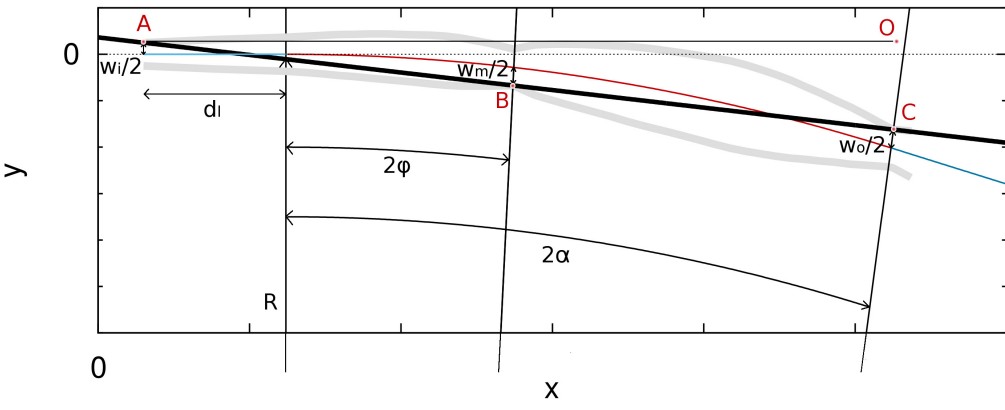

**Figure 3.** Drawing of the LOS condition. All points, distances and angles are explained in the text.

Feeding numbers into Equation (2) and (3), we plotted $w_m$ and $w_o$ for different $2\phi$ values, i.e., different locations of point B (Figure 4) We expected the region of interest to be where $w_o$ was slightly larger than $w_m$. Equation (2) and (3) gave us the limit for the LOS. In reality, this was not yet satisfactory, because without sufficient shielding, neutrons could enter the guide again. Therefore, the width at point B was chosen to be even smaller, namely by 5 mm.

The parameter to optimize was then $2\phi$, and the LOS condition was assured by setting $w_m$ and $w_o$ according to Equation (2) and (3) via $2\phi$. Practically, $2\phi$ could take any number, and we set it such that one of the predefined guide pieces was split. As we can see, the ROI was quite narrow (0.01° corresponds to about 0.25 m), and the optimum solution fell well within one foreseen 0.5 m long guide piece.

Equations (2) and (3) were derived for our specific case, with a given guide path and geometrical constraints, but the same principle can be applied to any guide geometry.

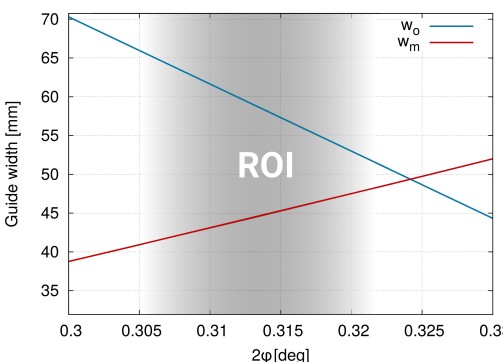

**Figure 4.** Guide widths that fulfill the LOS condition for different values of 2*φ* (location of point B). The estimated Range Of Interest (ROI) is marked in yellow.

### 5.3. Guide Shape Optimization

To optimize the guide shape, we first used guide_bot to find the maximum gain and determine which guide geometry performed, in general, the best. According to our boundary conditions, the guide entry and exit size and guide length were the constraints. First, basic combinations were investigated without the LOS condition. In a more advanced state of the optimization process, the LOS condition was requested to be fulfilled within the neutron guide bunker. Many combinations of straight, curved, elliptic, and parabolic shapes (including kinks) with an increasing number of guide sections were tested. The straight guide pieces were used in the reference guide (in combination with a curved one) and, later, with an increasing number of components, also as construction elements for a polygonal shape. It turned out that in our case, the elliptical form (alone or in sections) performed better than the parabolic form. A simple ellipse with a straight path led to an average gain of almost 4 across the entire wavelength range. In this case, the entry and exit dimensions had to be maximal. The achievable gain was smaller, but still significant, when the boundary conditions (especially the preservation of the original path and the fulfilment of the LOS condition) were taken into account.

In the following, the guide shape optimization process, taking all geometrical boundary conditions into account, is described. From here on, Mcoptimise was used instead of guide_bot. We followed a step-by-step optimization process to keep the number of parameters at a reasonable level. First, the shape was optimized with the best available supermirror coating, $m = 7$ (for the insert, $m$ was restricted to 4.5), with a maximum reflectivity of $R_0 = 0.995$ and an average slope of reflectivity of $\alpha = 3.78$ for all $m$ values. This slope corresponded to a reflectivity of 80% at about $m = 3.5$, matching the coating capabilities at PSI. Additionally, simulations were also run at lower $m$ values to explore the impact of the coating.

We started with the vertical shape, which was quite straightforward. Ideally, a simple ellipse would be chosen with a maximum entry and exit size, leaving the guide height as the only parameter to optimize. We also took our boundary condition of restricted space in the insert (first 4.6 m) into account. We plotted the intensity gain in comparison to the "half-ellipse" geometry, where the maximal guide height was equal to the height at the entrance, i.e., 12 cm (Figure 5). For the simulations, a constant guide width of 5 cm was used. Fortunately, the maximum allowed height of 25 cm (see the boundary conditions) was close to optimal. For this height, the linearly tapered insert (opened to 15.7 cm) fit well to the elliptical shape (Figure 6, top)and did not cause any severe limitation. At this point, we remark that the positions of the two gaps (0.26 m for the main shutter + beryllium filter and 0.11 m for the experimental shutter) were chosen based on space availability, and their presence was negligible in terms of guide performance. We ended up with the following vertical shape: (i) a linearly tapered insert (length: 4.6 m, height increasing from 12 to 15.7 cm), followed by (ii) an ellipse (length: 63.3 m, max. height: 25 cm). We also remark that the focal distances were not only wavelength-dependent but also strongly influenced

by the divergence (Appendix A). Therefore, we do not recommend to design a guide based on theoretical focal lengths, unless corrections for beam divergence are taken into account.

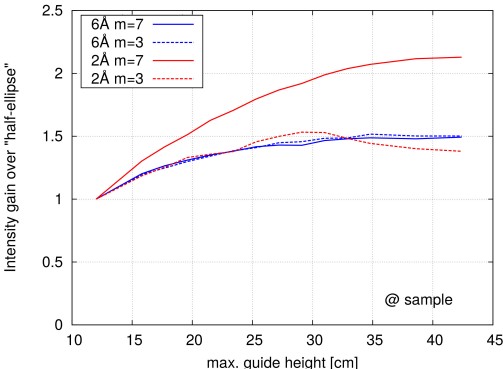

**Figure 5.** Effect of the maximum height of the elliptical guide on the intensity at the sample (Mc-Stas simulation).

After fixing the vertical shape of the guide, the horizontal geometry was addressed. Here, we had a geometrical limitation in the insert part and a restriction based on the above-derived LOS condition, and the guide had to follow the original curved path. According to the preliminary simulations, the guide shape after the curved section should be a simple ellipse. Its end size was fixed according to the boundary conditions, and the entry size was expressed with $2\phi$ according to the LOS condition. The only free parameter of the ellipse (apart from $2\phi$) was the maximal guide width.

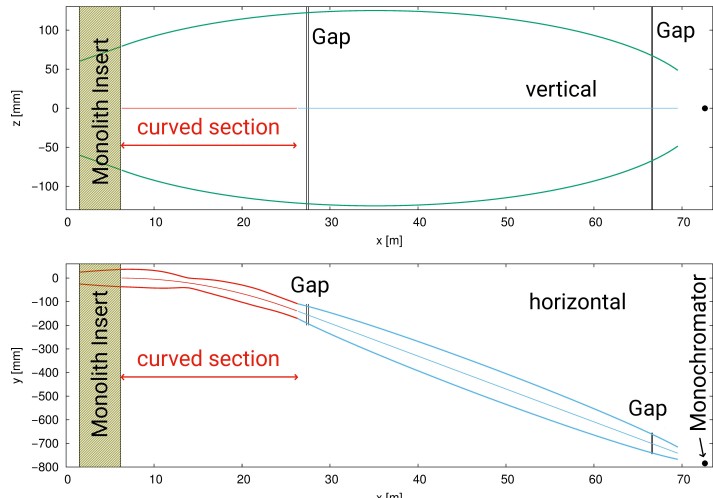

**Figure 6.** Geometry of the new guide. The green and blue parts are ellipses, and the red part has a free-style "peanut" shape, which fulfills the analytical LOS condition. The insert section has some geometrical limitations. The guide is constructed from 0.5 m long straight segments.

For the insert and curved section, we expected a "peanut" shape to be optimal, i.e., to have an increased guide width whenever possible (ballistic), with restricted widths at points B (at the $2\phi$ position) and at the end of the curved section (point C). Here, we did not use a specific geometrical form; instead, the whole geometry was optimized freely. The curved section was composed of alternating sequences of linearly tapered guide pieces and kinks. The optimization was performed in an iterative way because of the large number of free parameters and the broad range of possible values. In the beginning, the optimization was carried out for 2, 4, and 6 Å separately. For higher wavelengths, the guide was slightly broader, because of the higher critical angle of reflection. Nevertheless, cross-simulating the optimal shapes with the other wavelengths showed intensity differences within a few %

only. Therefore, detailed optimization was carried out at the middle of the wavelength range, at 4 Å. To speed up the simulation, the choppers were removed. We optimized the intensity at the sample position for the standard sample size of $15 \times 50$ mm$^2$, but, as discussed before, this was equivalent to a maximum intensity at the monochromator or any other sample size. First, the guide was divided into 2 m long pieces to find a rough shape; later, it was further divided into shorter sequences with a minimum length of 0.5 m. We also narrowed the boundaries for each value step-wise, according to the previous best values. Essentially, this was a brute-force method, with some manual help for convergence. It turned out that the intensity was not sensitive enough to variations in the guide width of less than 1 mm ($\leq 0.1$ deg change). Therefore, finally, the guide shape was smoothed. We found during the particle swarm optimization that a $\pm 5\%$ variation in the guide width resulted in up to 15% less intensity. Furthermore, the section after the "neck" had less impact on the performance than the section before the neck. The bottom of Figure 6 shows the resulting shape. The widest part of the guide was the middle of the ellipse, at 13 cm.

### 5.4. Guide Coating Optimization

In the next step, we optimized the coating of the guide. First, we simulated the guide for our characteristic wavelengths with a uniform coating possessing different *m* values either on the sides or on the top and the bottom (Figure 7). We kept m = 7 for the remaining walls (top/bottom or sides) and m = 4.5 for all sides of the insert section. As expected, shorter wavelengths were more strongly affected by changing the coating. We therefore performed the optimization for 2 Å. For the optimization, an in-house cost function was introduced, which took the sputtering efforts into account. Several optimizations were performed for differently weighted intensities relative to the cost The 0.5 m long guide pieces were grouped first based on their orientation (angle) relative to the guide axis, and for each group, a common m was obtained. Based on the result, the grouping was improved in an iterative manner. We optimized first the vertical, then the horizontal coating, the latter in two steps. In the second step, the horizontal coating was changed only for the curved section by allowing different values for the inner and outer side (Figure 8). In practice, the final *m* values had to be restricted to half-integer numbers, for which coating "recipes" exist. Based on the optimized values, several combinations of rounded m values were simulated. The best combination was selected based on the intensity (cost) relation (Figure 8). In the vertical plane, we ended up with 95% of the intensity for 15% of the cost in comparison to the m = 7 coating, and in the horizontal plane 98% of the intensity for 10% of the cost. For the same price, a uniform coating with m = 3.3 (2.7) could be realized, but this would give only 81% ( 63%) of the intensity.

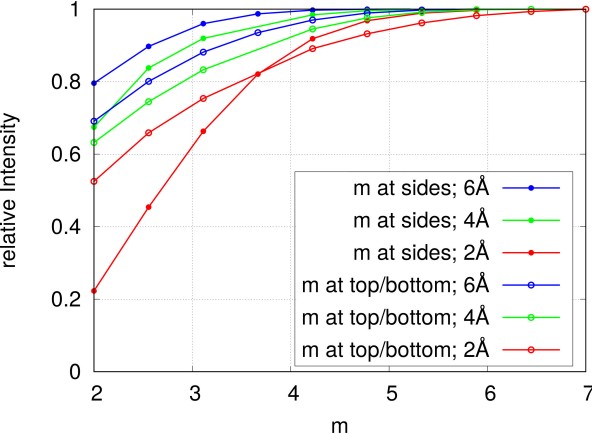

**Figure 7.** Effect of coating (m) on the intensity for different wavelengths (m was changed only for the given sides of the guide).

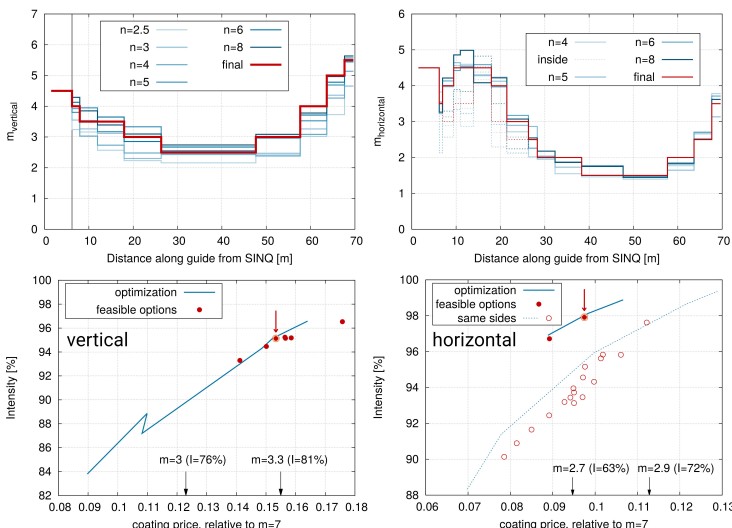

**Figure 8.** Top: Optimized m values (blue) along the guide for the vertical (left) and horizontal (right) planes for differently weighted intensities relative to the cost (($intensity)^n / cost$). The realized coating is plotted in red. Bottom: The corresponding intensities as a function of cost. The cost for a uniform coating with m = 7 was taken as 1. The outcome of the optimization is shown by the blue line, whereas feasible options (with m as a half-integer) are marked with red solid circles. Additionally, red arrows show the realized coatings. The dotted line and the open circles are cases of equal coatings on the inner and outer sides. For comparison, some values for uniform coatings of the guide are indicated by black arrows and corresponding numbers.

### 5.5. Performance

First, we compare the changes in the transported divergence and illumination at the monochromator (at ≈72.6 m away from the source and 3 m away from the end of the guide) obtained by the simulations (Figure 2, right- vs. left-hand side). The new guide clearly met our goal: the transported divergence was significantly increased, which resulted in a larger area of illumination. Actually, the original size of the monochromator fit the new guide quite well and transported the maximal possible divergence using state-of-the-art technology. The beam profile was more structured than before, because of the varying cross-section of the guide. Nevertheless, the beam profile at the sample remained homogeneous since it was smeared out by the mosaicity of the monochromator.

The white beam flux directly after the Monolith Insert (at 6.1 m from the source) remained essentially the same, based on gold foil activation measurements (Table 1). The neutron current in the guide, on the other hand, was increased by 1.9 times, since the guide cross-section was increased by this amount. The tapered shape and better supermirror coating of the insert guide led to this increase in the total number of neutrons in the guide.

The white beam flux gains measured at the end of the guide (≈69.6 m from the source) were significantly higher (Table 1) because of the more efficient transport/reduced losses due to the wide-cross-section elliptic guide and better supermirror coating. We determined from both the simulation and monochromatic flux measurements (at the sample position) that the spectrum was shifted towards shorter wavelengths, i.e., the gain was higher at shorter wavelengths (see the next paragraph). Both detection methods were based on a neutron absorption process, where the detection efficiency was proportional to the wavelength (higher absorption probability at longer wavelengths). This meant that the actual increase in the white beam flux was higher than the numbers in the table, which were obtained directly from the activity or counting ratios. The difference between the monitor and gold foil intensity gain was mainly caused by the accuracy of the gold foil measurements, which was about 15%, and by the respective changes in their sensitivity. The intensity distribution across the guide area (dysprosium sheet measurements, Figure 9) was found to be quite uniform. We estimated that the neutron flux at the gold foil position

was no more than 2% below the average value, i.e., it explained only a small part of the discrepancy.

**Table 1.** White beam intensity gain factors relative to the old guide. Flux refers to the measured area, while neutron current is the gain over the entire guide cross-section.

| Method | Position | Measured Area | Flux | Neutron Current |
|---|---|---|---|---|
| Gold foil | After Monolith Insert | $\varnothing$ = 1 cm | 1.0 | 1.9 |
| Gold foil | At guide end | $\varnothing$ = 1 cm | 2.5 | 2.5 |
| Fission chamber monitor | At guide end | 5 cm × 9.5 cm | 2.8 | 2.8 |

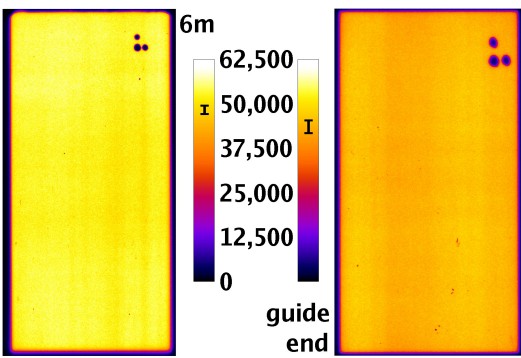

**Figure 9.** Beam profile measured by dysprosium foil activation after the guide upgrade. Left: at 6 m position (after the insert). Right: at the guide end. Please note that the guide cross-sections at 6m and at the guide end were 7.33 × 15.715 cm$^2$ and 5.0 × 9.5 cm$^2$, respectively, i.e., the pictures are scaled differently. The intensity variations are indicated by the black bars on the respective color scale bars. The values itself are arbitrary and should not be compared (they do not reflect the transport efficiency of the guide).

The monochromatic beam flux intensity gains for different wavelengths at the sample positions compared to the old guide are listed in Table 2. The gain for long wavelengths came from the reduced number of reflections due to the ballistic shape. At short wavelengths, the better coating was the dominating factor. At 6 Å the measured gain (vanadium) was slightly higher than the simulated value, because of degradation of the old guide, which was observed by gold foil monitoring before the upgrade. Assuming a lower m value in the insert for the degraded guide, the simulated gain factors could have been up to 50% higher. At short wavelengths, the measured gain was clearly lower than expected. The reason for this, at least partly, was the higher slope ($\alpha$) and non-linearity of our reflectivity curves , which we believe to be connected to the unusually thick glass substrates.

**Table 2.** Monochromatic intensity gain factors relative to the old guide. Simulated reference values refer to the non-degraded guide.

| Guide or Measurement Conditions | 2 Å (20 meV) | 4 Å (5.11 meV) | 6 Å (2.27 meV) |
|---|---|---|---|
| Old shape, m = 4.5 (insert)/m = 7 | 3.28 | 1.36 | 1.12 |
| New shape, m = 4.5 (insert)/m = 7 | 6.07 | 2.45 | 1.81 |
| New shape, m optimized | 5.52 | | |
| Measured, vanadium (in the detectors) | 4.0 | 2.1 | 1.9 |

The measured absolute values of the monochromatic neutron fluxes at the sample position are listed in Table 3. Very good agreement was found between the Au foil and $^3$He measurements. The monitor values had some uncertainties in terms of counting efficiency, but the overall agreement was good.

**Table 3.** Measured absolute values of the new monochromatic neutron fluxes (n/cm$^2$/s) at or close to the sample position, taking an average proton current of 1.7 mA at SINQ. Please note that the provided numbers already took a typical chopper ratio into account, i.e., only 1 out of N neutron pulses were used. This ratio N was 1, 2, and 3 for setups 3, 4.3, and 6 Å, respectively. The measurement errors were typically below 5%. The monitor was located behind the sample slit, which was fully open and covered the full beam area.

| Method | Measured Area | 3 Å (9.09 meV) | 4.3 Å | 6 Å (2.27 meV) |
|---|---|---|---|---|
| $^3$He detector | $\varnothing$ = 1 mm | $1.84 \times 10^5$ | $6.32 \times 10^4$ | $1.01 \times 10^4$ |
| Gold foil | $\varnothing$ = 10 mm | $1.73 \times 10^5$ | $6.17 \times 10^4$ | |
| Fission chamber/monitor | 2.7× 5 cm$^2$ | $1.63 \times 10^5$ | $5.13 \times 10^4$ | $1.13 \times 10^4$ |

Finally, we compare the measured time-of-flight spectra of the vanadium standard (Figure 10), using an incoming neutron wavelength of 4.3 Å. Both the energy resolution and the background (normalized to the incoming neutron flux) were essentially unchanged. The first one was not a surprise but expected. The latter one was a much more complex question, and therefore less easy to predict. At short incident wavelengths, below the aluminium edge of 4 Å, the background is dominated by neutrons scattered on the sample environment devices, like magnets, which are often used. Therefore we focus here on the performance at longer wavelengths. Here, a major contribution to the background arose from cold (or thermal) neutrons that were reflected by the monochromator but did not hit the sample. Therefore, the background was expected to scale roughly with the monochromatic neutron flux. The new guide illuminated the monochromator better, which resulted in a higher amount of neutrons passing beside the monochromator and may have created a background. This, together with the shift of the spectral weight to shorter neutron wavelengths, might have slightly increased the background. On the other hand, the guide was now more than out of the LOS. It seemed that none of these minor changes had a measurable net impact on the background.

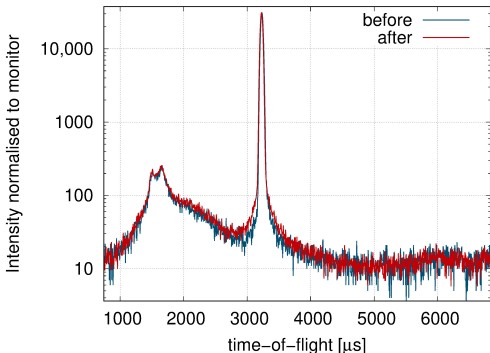

**Figure 10.** Comparison of standard vanadium measurements ($\lambda = 4.3$ Å) before and after the guide upgrade. The spectra were normalized, as usual, to the monitor in front of the sample.

Since the upgrade, experiments at higher energies (between 1 and 2 , using the weaker (004) reflection) have also been possible. We conducted a successful user experiment with $\lambda_i = 1.3$Å. Here, we could measure excitations up to 40 meV (at the neutron energy loss side), which was not feasible before due to the very low flux. Since FOCUS is the only spectrometer for powder (or liquid) samples at PSI, this extended capability is of high importance.

## 6. Summary and Conclusions

The design and performance of the new ballistic guide for the time-of-flight spectrometer FOCUS at PSI are presented here. It is a 68 m long guide with a large cross-section.

The aim of the new guide was to increase the neutron flux at the sample without changing the instrument performance and remaining within the geometrical constraints given by the building and neighboring instruments. Furthermore, the direct line of sight (LOS) was requested to end within the neutron bunker in order to minimize the background in the neutron hall.

We achieved a significant flux increase: by a factor of 4 to 2 in the wavelength range of 6 to 2 Å, independent of sample size. The new guide had an improved supermirror coating and a dedicated ballistic shape in order to increase the transported phase space and reduce the losses via fewer reflections. The energy resolution was maintained by fixing the guide end width. We derived an analytical form for the LOS condition, which is adaptable to any other guide geometry and is convenient to use in automated optimization processes. The final design was composed of elliptical parts and a "peanut" shape. The latter emerged from a free optimization of the shape in the horizontal plane for the insert and the curved section of the guide. The parameters of the ellipses were also computer-optimized since we observed that the focusing properties strongly deviated from the ideal case of a point source.

After the upgrade, experiments became faster, and boundaries were pushed further, e.g., higher incident energies ($\approx$50 meV) became possible. In the future, we hope to further increase the flux and the signal-to-noise ratio by, e.g., upgrading the monochromator.

**Author Contributions:** Conceptualisation, design, simulations, evaluation, test experiments, flux measurements, writing, visualization, review, and editing, F.J.; simulations, flux measurements, review, and editing, M.Y.; flux measurements, supermirror coating, review, and editing, C.K.; lead engineer, L.H.; project lead, resources, review, and editing, U.F. All authors have read and agreed to the published version of the manuscript.

**Funding:** Shielding design for the whole SINQ Upgrade project was partly financed by SNF, Grant 200021_150048/1; the rest was financed by PSI internally.

**Data Availability Statement:** The data presented in this study are available on request from the corresponding authors.

**Acknowledgments:** This project would not have been possible without numerous contributions. Software: Markus Appel (ILL) and Mads Bertelsen (ESS). Beamtime: ISIS/Robert Dalgliesh, ILL/Thomas Saerbeck, and LLB/Frédéric Ott. Supermirror coating: Michael Horisberger, Michael Blumer, Stefan Fischer, Francesco Guarascio, Manuel Perrass, and Roger Stefani. Scientific and technical discussion quality management and realization: Marek Bartkowiak, Roman Bürge, Jan Peter Embs, Artur Glavic, Christian Kägi, Peter Keller, Joachim Kohlbrecher, Peter Ming, Thomas Mühlebach, Christof Niedermayer, Emmanouela Rantsiou, Jochen Stahn, and the whole SINQ Upgrade team at PSI.

**Conflicts of Interest:** The authors declare no conflicts of interest.

## Abbreviations

The following abbreviations are used in this manuscript:

| | |
|---|---|
| LOS | Direct line of sight |
| ROI | Range of interest |
| PG | Pyrolytic graphite |

## Appendix A. Focusing Properties

The focusing of a neutron guide with a large cross-section that transports high divergence is altered significantly, i.e., the beam size minimum can be positioned far from the theoretical focal point. To illustrate this, we simulated the beam profile after a simplified version of the FOCUS guide. It had no curved part, the vertical shape was a simple ellipse, and it was straight horizontally, i.e., it had a constant width. We placed at every 0.5 m a position-sensitive detector until the monochromator position was reached. We plotted the results for a neutron wavelength of 2 Å, and we could tune the transported divergence by choosing (realistic) m values of 3 and 7 (Figure A1). We observed that divergence

significantly altered the focusing behavior. In the case of 6 Å and an m = 7 coating (not plotted), the divergence dominated, and the beam size increased continuously.

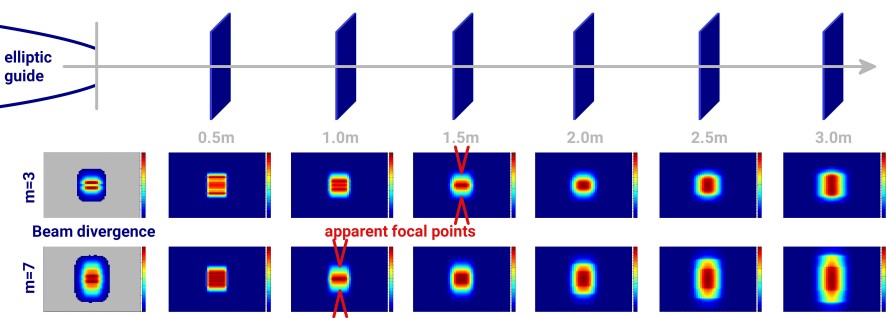

**Figure A1.** The beam divergence shifts the apparent focal point of the guide. Here, the beam profiles are shown after a simplified guide at several distances for $\lambda_i$ = 2 Å. The first picture in each row shows the divergence at the end of the guide.

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
