# Peer review of "New Ballistic Neutron Guide for the Time-of-Flight Spectrometer FOCUS at PSI"

_qubs, doi:10.3390/qubs8010008_

Round 1

Reviewer 1 Report

Comments and Suggestions for Authors

This paper presents the design and performance of the new ballistic guide for the inelastic neutron scattering spectrometer FOCUS at PSI. The aim is to increase the flux significantly while maintaining energy resolution, making this excellent instrument better. The design is strongly supported by the convincing simulation results. The work will greatly benefit the neutron scattering instrumentation. 

Therefore, I recommend the paper to be accepted for publication in its current form.  

Author Response

We are honored to obtain such a very positive review, thanks a lot!

Reviewer 2 Report

Comments and Suggestions for Authors

I found the topic extremely interesting and the article well written.

I just have a few minor comments/suggestions:

- Are the results presented in Figure_1 the outcome of measurements or of computation? I feel the caption should stated that.

- Figure_2 think about having units and scale (numbers) for the vertical axis. I know it is just a sketch to exemplify, illustrate, the LOS condition. I leave the final decision to your judgment.

- Figure_3: may be using for the ROI a color different from "yellow" can improve readability in all the situation (depending of the display settings or in grayscale, it results very faint)

- Figure_4: use Angstrom symbol in the key instead of AA. 

- Figure_5: I found weird the vertical axis is named "y" for both vertical and horizontal cases. Personally I'd have used "z" for the beam direction, but going in that way would mean redo all the figure, which I don't suggest (and even less I'm going to require), therefore, what's about considering "z" for the horizontal. Axis label of Fig2 should be changed coherently. 

- Figure_6: use Angstrom symbol in the key instead of AA. 

- Figure_7: bottom-right: the key for open circle is missing

- Figure_7: bottom (both): make the labels "vertical" and "horizontal" bigger

- Figure_7: bottom (both): use of a German word (preis) in the horizontal axis title

- Figure_7: caption: bigger points = "solid circles", circle =" open circle" ?

- Figure_8: caption: it is not clear to me what does it mean "The intensity variation is normalized and indicated on the respective color scale bars." Is the scale for the two plots the same? What are the units? Arbitrary Units?

Reviewer 3 Report

Comments and Suggestions for Authors

The manuscript reports on the new ballistic neutron guide for the time-of-flight spectrometer FOCUS at PSI. A significant intensity increase in the neutron flux was achieved. The methods of obtaining this, as well convincing experimental results are presented. The topic is of interest, the manuscript is well written, therefore it should be published.

There are few places where the authors should check the manuscript, and make modifications as necessary.

-lines 399-400: one could reword the gain of 1.0 to something saying that there was not change (not important, it just reads weird)

-lines 409-412: there is a sentence here which needs to be rewarded.

-line ~414 and figure 8: the text speaks about gold foil monitor, but the caption of figure tells us that dysprosium foil activation was used.

-line 436: it should be Fig.9. (not Fig. 8.)

-figure 7, the two panels at the bottom: preis is in German, the authors might want to change it.

Reviewer 4 Report

Comments and Suggestions for Authors

The manuscript presents results of calculated and implemented modification of the neutron guide for the time-of-flight direct geometry spectrometer FOCUS at PSI. The task was to make upgrade preserving the current geometrical constraints, double focusing monochromator, etc. Different shapes and coating values of the guide in vertical and horizontal directions were considered to optimize the efficiency and the cost of the new guide. The authors show that the new guide has almost the same performance for energy resolution and background, and provide significant increase of delivered neutrons at the sample position, factor 4 to 2 for the neutrons with wavelength range of 6 to 2 A.

As a comment, I would like to note that for easy reading of the manuscript, it would be nice if the authors place additional figure in the Introduction to show the schematic view of the FOCUS spectrometer with the positions not only the neutron guides, but also the monochromator, Fermi chopper, sample, and the detectors. Otherwise, the readers need to search the earlier publications on FOCUS spectrometer to understand the details.

To conclude, my recommendation is the manuscript deserves to be published in the Quantum Beam Science.

Author Response

Thank you for the suggestion, it is indeed useful. We added the sentence "See layout on Fig. 1." at the end of the first paragraph, included the new Fig. 1, renumerated all other figures and references to them, and moved Fig. positions in some cases to avoid big empty spaces.